# How positive deviants helped in fighting the early phase of COVID-19 pandemic? A qualitative study exploring the roles of frontline health workers in Nepal

**Rolina Dhital**[1]*, **Madhusudan Subedi**[2], **Pawan Kumar Hamal**[3], **Carmina Shrestha**[1], **Sandesh Bhusal**[1], **Reshika Rimal**[1], **Lovin Gopali**[1], **Richa Shah**[1]

1 Health Action and Research, Kathmandu, Nepal, 2 Patan Academy of Health Sciences, Lalitpur, Nepal, 3 National Academy for Medical Sciences, Kathmandu, Nepal

* rolina.dhital@gmail.com

**Data Availability Statement:** Underlying data is available within the manuscript itself and in supplementary files.

## Abstract

Positive deviance is an approach wherein learnings from persons who fare better than their peers under similar circumstances are used to enable behavioral and social change. Such behaviors and solutions are likely affordable, acceptable, sustainable, and fit into the socio-cultural milieu. Despite the wide use of positive deviance in many public health programs and research, it has yet to be used to study frontline workers in the context of COVID-19. Therefore, this study aimed to explore the positive deviance traits among frontline health workers during the early days of the COVID-19 pandemic in Nepal. This qualitative study followed a grounded theory approach. The data was collected through in-depth interviews among the 17 identified participants representing different cadres of the health workforce, types of health facilities, and regions across Nepal purposively. The findings are structured around four major themes: challenges, finding solutions and innovations, positive lessons, and motivations. The personal challenges included fear and anxiety about the uncertainties. The professional challenges included stigma, infection control, and changing work style with the use of personal protective equipment. Despite the challenges, they managed available resources and innovated low-cost, technological, and practice-based solutions. They were able to reflect upon the positive lessons learned, such as self-sustainability, teamwork, and policy direction and research, and self-reflection of personal growth and patient care. The intrinsic motivation included their inherent value system, and the extrinsic motivation included appreciation and acknowledgment, family and social support, psychosocial support from peers, and support from higher authorities. This study provides insights into how the positive deviance approach can help identify the solution amid the most challenging circumstances, such as the COVID-19 pandemic in low-resource settings. However, more extensive studies are warranted to explore deeper into positive deviance and its long-term effects in bringing positive outcomes during the pandemic.

**Funding:** The authors received no specific funding for this work.

**Competing interests:** The authors have declared that no competing interests exist.

## Introduction

Positive deviance is a concept widely used in public health and social sciences. It can be considered a practical strategy that enables behavior and social changes by learning from the deviants who fare better than others under similar circumstances [1, 2]. In any setting, there are people whose uncommon but successful behaviors or strategies enable them to find better solutions to a problem than their peers, despite facing similar challenges and resource constraints. Such behaviors are likely affordable, acceptable, sustainable, and most socially and culturally contextual [3]. Such positive deviances could also have helped identify sustainable solutions to tackle the COVID-19 pandemic amid resource constraints [4].

The positive deviance approach has been proven successful in identifying sustainable solutions, especially for resource constraint settings. It has been widely used in various public health programs related to nutrition, maternal and newborn health, sexual and reproductive health, and HIV/AIDS [5–7]. The positive deviance approach has also been successfully implemented in healthcare settings to prevent healthcare-associated infections [8], reduce medication errors [9], and improve hand hygiene practices [10]. A systematic review that explored the positive deviance approaches within the primary care settings identified that positive deviants were more successful in improving disease management and health promotion, and positively influenced patients' health-related behaviors [11]. A review of nurses' role in positive deviance indicated that clinical settings, even in general circumstances, comprise uncertainties, and the decision-making often involves risks as the applicable professional standards may not always be readily available [12]. Under those uncertain circumstances, the nurses working at the frontlines and those able to adapt and act differently to changing circumstances were more intuitive in realizing what could work beyond the norms than the higher authorities who were not working at the frontlines [12].

Given the usefulness of the positive deviance approach in understanding and enabling change in the health sector, the approach can be used in the context of COVID-19. Amid the lack of resources and numerous challenges, positive deviance would allow us to look into the successful examples among frontline health workers in resource constraint settings. Frontline health workers comprise different cadres of the health workforce, such as specialist doctors, medical officers, nurses, midwives, lab technicians, and more, who provide care directly in the hospital or to their communities [13]. Despite the crisis and difficult circumstances faced widely by frontline health workers across the globe due to COVID-19, many were performing better and were more successful in preventing the outbreak and treating COVID-19 patients amid the resource constraints [14, 15]. Better knowledge and skills, better planning and management, higher levels of inbuilt compassion and motivation, and better ideas and innovations were some key qualities identified among these health workers in previous studies that enabled them to perform better [14–16]. Such outliers among the health workers who are finding better solutions amid the crisis could also be considered positive deviants; however, their roles through the lens of positive deviance have not been explored adequately.

Nepal, as a newly emerged lower-middle-income country, faced a myriad of challenges in addressing COVID-19 during the early stage of 2020 due to resource constraints [17]. Moreover, the reshuffled health system under the newly formed federal Nepal may not have been adequately equipped to tackle the unforeseen circumstances [18]. The weaker health system capacity in terms of finance, medical products, technology, and human resources for health has created extreme pressure on health workers to provide appropriate diagnostic and treatment services [19]. It has led to increased work burden and stress among healthcare workers who are the front liners tackling various challenges during times of crisis [20].

Exploring the roles of frontline health workers through the lens of positive deviance shall provide a fresh perspective and positive outlook on the COVID-19 response in low-resource

countries such as Nepal. Understanding their experiences and perspectives could provide us an opportunity to look into locally available solutions to tackle similar problems. Such positive examples and experiences could be sustainable solutions that could be scaled up and expanded within the country as well as to countries with similar settings. Therefore, this study aimed to explore the experiences of frontline workers during the early days of the pandemic and identify the key examples of positive deviance with a specific focus on the earlier days of the pandemic during the first wave in Nepal.

## Materials and methods

### Study design and settings

This qualitative study explored the positive deviance traits of the health workers fighting COVID-19 in Nepal through the grounded theory approach [21]. We followed the Consolidated Criteria for Reporting Qualitative Studies (COREQ) guideline to conduct, analyze and report the findings of this study [22]. We aimed to generate positive traits and insights that explain the action in a social context. Our conceptual framework was developed from the data rather than from previous studies [21].

We included health workers working in different layers of the health system, starting from the primary health care level, district hospital level, and provincial and central hospital levels across Nepal. We purposively selected the health workers representing different cadres, and types of health facilities across Nepal.

### Steps of positive deviance approach

There are different proposed methodologies of positive deviance approaches. However, a common ground for all the methodologies is identifying the positive traits of individuals in the community and identifying the solutions for positive changes [23]. A common framework used for positive deviance approaches, particularly in healthcare settings, includes four major steps. Step 1 is identifying positive deviants who demonstrate consistent, unparalleled performance in an area of interest. Step 2 is where qualitative methods are used to study the practices of positive deviants in-depth and generate hypotheses. Step 3 comprises testing the hypothesis in larger representative samples. The last step, Step 4, is to share evidence about best practices in collaboration with stakeholders and potential adopters [24, 25].

This study focuses on the first two initial steps of the positive deviance approach, which includes identifying the positive deviants and studying their positive traits and practices at a deeper level to understand how such traits could have helped in the earlier phases of the COVID-19 pandemic. The latter steps of the positive deviance approach (Steps 3 and 4) would warrant a separate follow-up study to provide a long-term perspective and thus were beyond the scope of this qualitative study. Fig 1 summarizes the synthesized flow of the study based on the steps of positive deviance used in this study and how it may pave the way for future research to be tested in a larger population.

### Identification and selection of positive deviants

We created a list of health workers with potential positive deviance traits based on the information about their work available in public media and through recommendations by the stakeholders aware of such frontline health workers. We also consulted public health researchers, frontline health workers, health institutions, and identified their roles and activities mentioned and appreciated on social media. The positive deviance traits focused on four conditions 1) actively working as frontline health workers during the early phase of the

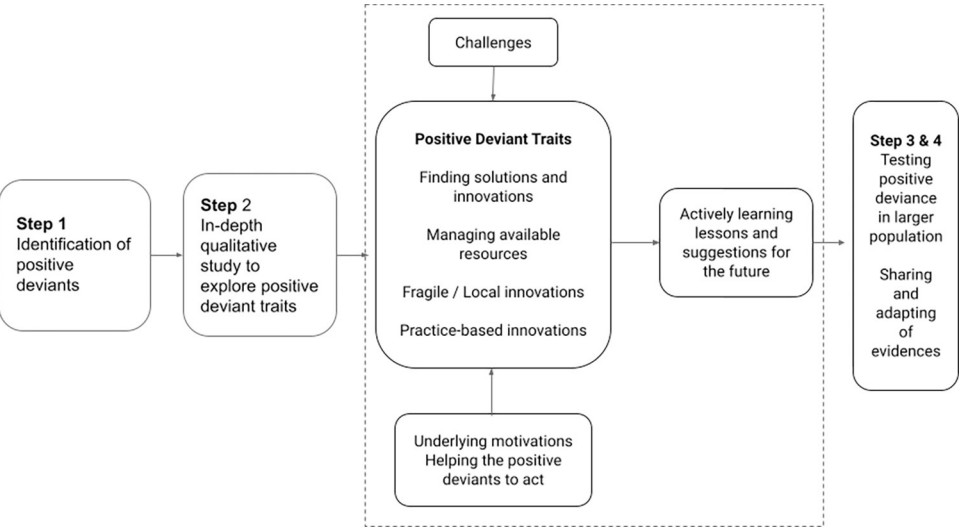

**Fig 1. The flow of the study based on the steps of the positive deviance approach.**

COVID-19 pandemic, 2) successfully managing to overcome the challenges in their management despite the limited resources, 3) innovating unique mechanisms to overcome the challenges during their management, and 4) managed to improve the overall management within their working areas.

We then contacted them via a telephone call, followed by an email inviting them to attend the video call via zoom before the interview. All health workers approached consented to participate in the interview, and there was no refusal to participate. We collected data until the information on their experiences reached the point of saturation.

## Data collection

We collected data through in-depth interviews using an interview guide. Considering the pandemic and the need to maintain safety precautions, we conducted all interviews through online video calls using the zoom platform. We used an interview checklist that explored the positive deviant traits such as their backgrounds, perceptions, challenges, ways of overcoming them, factors motivating them to work, innovations they have identified, and their suggestions to tackle the pandemic in Nepal (S1 Table).

The qualified and experienced researchers (RD, MS, and PKH) conducted interviews. PKH, a qualified male clinician, was primarily responsible for coordinating and scheduling the interviews for the participants. MS, a male professor, and an experienced qualitative researcher, ensured the interviews were conducted following COREQ guidelines and checklist (S2 Table). RD, MS, and PKH all established a good rapport with the participants before initiating the interviews to keep them at ease in sharing their experiences. All three researchers introduced themselves and clearly explained the objectives to the participants in every interview. RD, an experienced female public health researcher, conducted most of the interviews, and MS and PKH facilitated and took notes during the interviews.

We conducted video-recorded interviews based on the participant's preferred time, where the participant could participate in the interview without interference from other people. We maintained privacy where only the interviewer, note takers, and the interviewee attended the online video interview. We ensured confidentiality by maintaining the anonymity of the participants, with the interview data being accessible only to the research team.

The interviews lasted for 1–1.5 hours. Following the interviews, the researchers discussed the key points with the participants to ensure they understood what they were trying to explain. The interviews were in everyday conversational language with a mixture of Nepali and English. The interviews took place between 8 November 2020 and 16 March 2021, with no repeat interviews.

### Ethical considerations

We obtained ethical approval for this study from Nepal Health Research Council (regd. 600/2020). We sent an invitation email to all the participants with the information sheet and consent form. We requested them to go through the information sheet and informed consent and reply if they were willing to participate in the study. We obtained the video-recorded informed consent before the interview, where the participants declared that they had read through the information sheet and were willing to participate in the study.

### Data analysis

The researchers (RS, CS, SB, LG, and RR) transcribed all the interviews from the recorded videos into Microsoft Word. We translated the Nepali expressions at the time of transcription. RS and CS ensured the quality of the translation and managed the data. RD, MS, and PKH ensured the overall quality of the transcripts. We then imported the transcribed data to Dedoose 9.0.17, a web application for managing, analyzing, and presenting qualitative findings [26].

All the researchers read and re-read each transcript to make sense of the content and meanings. We first created the root codes and grouped the codes with similar meanings into categories. We then grouped the categories with similar meanings into four major themes (Table 1). MS supervised the overall data analysis process and finalized the codes, categories, and themes generated after discussing with RD and all the researchers.

## Results

### Characteristics of participants

Table 2 summarizes the general characteristics of the participants. In total, 17 frontline health workers working during COVID-19 pandemic participated in this study which included doctors, nurses, and a laboratory scientist. The majority of the participants were within the age group of 21–29 years of age, and the majority of the participants were married. None of the participants were infected with COVID-19 at the time of the interview though a few had recovered from mild COVID-19 symptoms before the interview took place. As the interviews were conducted at the earlier phase of the pandemic, none of the participants had been vaccinated against COVID-19 yet. Around the time of the interview, only the district, provincial, and central hospitals had been identified as the designated COVID-19 hospitals. The primary health centers were not yet designated as the COVID-19 treatment centers but were designated as the centers that managed health care for quarantine and isolation centers within their communities.

### Themes

The findings of this study are structured around four major themes, 1) Challenges, 2) Innovative solutions, 3) Motivations, and 4) Positive lessons (Table 1).

**1. Challenges.** Identifying the existing challenges is often the first step in the positive deviance approach to understand how the positive deviants were able to perform. In the earlier days of the pandemic, the vaccines against COVID-19 and the standard treatment protocols were still not in place. Most participants expressed that they had to rely on limited information

**Table 1. Themes, subthemes, and codes on positive deviance traits among frontline health workers during COVID-19.**

| | Themes | Subthemes | Codes |
|---|---|---|---|
| 1 | **Challenges** | 1.1 Personal challenges | Uncertainty, fear, anxiety, distance from family, stigma |
| | | 1.2 Professional challenges | Stigma from friends, colleagues at work |
| | | 1.2.1 Stigma | |
| | | 1.2.2 Changing working patterns | Increased working hours, increased workload |
| | | 1.2.3 Working with PPE sets | Working with PPE sets |
| 2 | **Finding solutions and innovations** | 2.1 Managing available resources | Utilization of resources, taking precautions, and encouraging their peers and colleagues |
| | | 2.2 Fragile innovations | Modifying, designing low-cost and local medical equipment |
| | | 2.3 Technological and practice-based innovations | improving lab tests, and treatment strategies, mobilizing communities, generating ideas |
| 3 | **Positive lessons** | 3.1 Professional lessons | |
| | | 3.1.1 Self-sustainability | Importance of local solutions, sustainable solutions, continuing with sustainability |
| | | 3.1.2 Teamwork | Peer support, team spirit, team management |
| | | 3.1.3 Policy direction and research | Lack of data, importance of evidence-based protocols/practices/guidelines, preparedness for future research |
| | | 3.2 Personal lessons | |
| | | 3.2.1 Self-reflection | Personal journey, personal skills, change in attitude/confidence |
| | | 3.2.2 Lesson from the patients | Empathy, compassion, patient care |
| 4 | **Motivations** | 4.1 Intrinsic motivation | Value of health worker, compassion, value, dedication, work ethics |
| | | 4.2 Extrinsic motivation | |
| | | 4.2.1 Appreciations and acknowledgments | Appreciations from society /communities/ patients, awards, rewards |
| | | 4.2.2 Family and social support | Encouragement from family/society, support system |
| | | 4.2.3 Psychosocial support for peers | Peer support programs, peer discussions, counseling |
| | | 4.2.4 Support from higher authorities | Support and coordination with higher authorities/local government |

and act amid uncertainties. Despite the fear and uncertainties and difficult working circumstances, their experiences of challenges reflected how they continued their work. The challenges faced by the participants as a result of the COVID-19 pandemic during the earlier days of the pandemic included personal and professional issues.

*1.1 Personal challenges*. Most participants reflected fears of the unknown, risks of getting transmitted and affecting their families, and the busy work schedule that made it difficult for them to manage their work-life balance.

*'I was very worried that I would infect my wife, my child, and the older people in my family. As I was treating COVID-19 patients very closely, I maintained a physical distance from my family members since the beginning of the outbreak. When COVID-19 started, my son was just six months old, and now he is 18 months old. I feel sad that I have missed his important milestones during this time. I was so consumed by work and worried about the patients all the time that I had to detach myself from my loved ones in the past year.'* -Male, 40s, urban, central hospital

*'We were supposed to leave our homes and live in a quarantined hotel during the week of our duty. I wrote a letter for my family just in case I wouldn't make it and never come back (says it while sobbing)'* -Female, 40s, urban, provincial hospital

*'My biggest fear was protecting my 82-year-old mother. She used to wait for me to have dinner together all the time. However, after I started working for COVID-19, I had to avoid spending physical time with her. '* - Male, 40s, urban, central hospital

**Table 2. Characteristics of the participants.**

| Characteristics of the participants | Number |
|---|---|
| **Age groups in years** | |
| 20–29 | 10 |
| 30–39 | 3 |
| 40 and above | 4 |
| **Gender** | |
| Female | 7 |
| Male | 10 |
| **Marital status** | |
| Unmarried | 5 |
| Married | 12 |
| **Profession** | |
| Doctors | 11 |
| Nurses | 5 |
| Laboratory scientists | 1 |
| **Location of facility** | |
| Urban | 11 |
| Rural | 6 |
| **Level of facility** | |
| Primary health care | 3 |
| District/Provincial hospital | 4 |
| Central hospitals | 10 |

*1.2 Professional challenges*. Some common professional challenges faced by frontline health workers in the initial days included stigma, challenges in infection control, and changing work style with the use of personal protective equipment sets (PPE).

*1.2.1 Stigma.* Some health workers felt stigmatized by relatives or their fellow health workers. Some expressed that they felt hurt and let down during those difficult times. The responses reflected the vulnerable circumstances of their work in the earlier days of the pandemic.

*'One of our colleagues had traveled to a remote community that was 15 hours away by foot from our hospital. One week later, when he returned, his landlord didn't allow him to enter the house where he was residing. So, we had to manage a hotel for him to stay in.'* -Male, 30s, rural, provincial hospital

*'I was also infected while treating the patients, and my colleagues called me "COVID-cha" (a casual way of saying that COVID-thing). They used to tell me not to come near them as I could infect them. They used to put on masks hastily and avoid me when they saw me, which did make me feel low.'* -Female, 50s, urban, central hospital

*1.2.2 Changing working patterns.* During the earlier days, many health workers were required to work extra hours due to the sudden surge of COVID-19 infections and the lack of adequate human resources. Increased working hours made the health workers burdened and exhausted with additional work.

*'During COVID, our duty hours extended to over 12 hours instead of normal. We were stressed and fearful of making mistakes with infection control and compromising quality care for the infected patients. We also had to be psychologically strong and minimize the anxiety,*

*and stress of staff to maintain good quality care to deal with the overburden of the work.'-* Female, 40s, urban, central hospital

**1.2.3 Working with PPE sets.** Many health workers shared the difficulties of working with PPE in the initial days of the pandemic. Some also expressed how it created a barrier for them with the patients. They felt they were not able to take care of or communicate with their patients as effectively as they would have been in other circumstances. At the same time, some had to wear PPE even when they were traveling to quarantine and isolation centers in the communities, which made it uncomfortable for them.

*'At the beginning of the pandemic, we used to travel to remote parts and screen migrant workers for COVID-19 based on their symptoms. We had to wear a PPE set when traveling to the communities. It used to be hot, and I used to feel thirsty but still wouldn't drink water the entire day so that I could avoid going to the toilet.'*—Female, 20s, rural, PHC

**2. Finding solutions and innovations.** Despite the different challenges, almost all the participants reflected that they were actively seeking solutions. They identified local solutions by managing available resources more efficiently and innovating new ideas. Their local solutions helped them manage patients and minimize the risk of infection without external support.

*2.1 Managing available resources.* Almost all health workers shared the plans and strategies that they utilized to control, manage, and treat COVID-19 patients within an under-resourced healthcare setting or catchment area. The health workers from higher centers, such as central hospitals, were able to manage laboratory-related resources and locally produce PPE sets. Whereas some health professionals from the provincial hospitals were better at mobilizing and coordinating human resources. Most of them were also able to encourage patients' involvement and community participation for infection control and disease management amid the lack of adequate health professionals in the hospitals. A few health professionals from the remotest parts of the country coordinated with the local governments and came up with solutions for infection control that were unique and effective to their local settings.

*'We also had a lot of problems regarding the lack of required instruments like the tubes to run laboratory tests (sealed pack, filtered, and autoclaved). Since there was a lockdown, the resources that we had in stock were being used, and we were not able to get new resources including new kits. We were lucky that we had equipment received via donation that we were able to use. When we were out of the PCR tube, I economized and used the existing tubes. We used to reserve PCR kits a month before.'*—Male,50s, urban, central hospital

*'When the rest of the country was struggling with a lack of PPE and had to go to the COVID-19 wards without adequate protection, we had already looked into solutions. We started producing our own PPE sets without wasting any time or being dependent on the government or others. Our staff started stitching the sets on a large scale which was enough for the health workers of our hospital. At a time when most hospitals could only afford to use 5 PPEs a day, we were using 30–40 PPEs per day for the safety of our hospital staff.'*- Female, 50s, urban, central hospital

*'There were around 20–21 COVID-19 patients on each floor, with 5–6 patients in each block. We nominated a group leader amongst the patients for every block. The leader was responsible for communicating with us on behalf of other patients and also had to inform us about the meal preference of their groups. They were also given the responsibility to conduct exercises*

*and yoga. We used to play music through the speakers, and the patients used to sing songs and dance. We created a positive environment for the patients.'*-Female, 40s, urban, provincial hospital

*2.2 Fragile innovations.* The health professionals from the remote areas, particularly the younger professionals working in the primary health centers, had to deal with more shortages of resources such as PPEs. However, the health workers working in the provincial and central hospitals also faced similar challenges. Many came up with fragile, frugal innovations born out of necessity amid a lack of resources. They used the locally available low-cost resources, which were not necessarily safe or the best options. Nevertheless, they identified the best solutions that they could come up with for the given circumstances.

*'Even though there was no proven evidence, we thought something would be better than nothing and decided to use raincoats as an alternative to PPE. We purchased raincoats and rain boots from the local market and even used utility gloves instead of surgical gloves when there was a shortage. We sterilized the raincoats, boots, and utility gloves every day. '*-Male, 20s, rural, PHC

*'I remember the farmers used to wear double-layered plastic to go to the field to protect themselves from pests. So, we just used that concept and created our own version of "PPE" with double-layered plastic. We used such plastic coverings for almost two weeks to manage suspected COVID-19 patients.'*–Male, 40s, rural, provincial hospital

*'We did not have the resources to separate the driver's seat and COVID-19 patient's compartment in the ambulances. The ambulance drivers were scared of getting infected by the patients they were carrying in their ambulance. At that time, we applied thick plywood to create a barrier between the two compartments. Later on, we started using plastic as a barrier.'*–Male, 30s, urban, central hospital

*2.3 Technological and practice-based innovations.* Most of the health professionals working in the central hospitals shared how they adapted to new practices and used technologies differently to overcome the new challenges during the pandemic. The solutions not only helped them control infection transmission but also in better communication for the patients.

***2.3.1 Technological innovations.*** Some health workers working in the central hospitals highlighted that they invented new low-cost technologies or promoted the use of existing technologies more efficiently to minimize the risk of infection from COVID-19 patients.

*'We used the technique of video laryngoscopy extensively during the procedure of intubation/ extubation, which wasn't used much before COVID-19; this might have lessened the risk of infection rate among health workers. To minimize the risk of infection while performing the aerosol-generating procedure, we designed a low-cost enclosure box (transparent plastic box) to intubate the patients.'*- Male, 30s, urban, central hospital

*'There was a huge challenge to collect the nasopharyngeal samples for the COVID-19 test. To address that challenge, local sample collection booths were prepared to collect the samples in the safest possible way.'*—Male, 40s, urban, central hospital

***2.3.2 Practice-based innovation.*** Many health workers shared that new ideas and solutions came to them through practices and experiences. Some health workers from the central hospitals shared how they modified their regular practices and came up with new ideas for infection control during surgery. A few health workers from the remote areas shared how they were able

to coordinate with the local government to come up with solutions to reach more people in a shorter time.

> *'We demonstrated that we could perform surgery safely on COVID-19 patients. We planned on how to perform surgery, place the operating system, decontaminate the fluid that comes out during suction, etc. These things helped in the safety of the surgical staff.'* -Male, 40s, urban, central hospital

> *'So many remote districts in our province are without road access. There are places where you have to walk for 4–5 days to reach. If those migrant returnees from India/other countries showed symptoms after reaching home, it was not possible to walk there to collect swabs as it would take 6–7 days. In this kind of situation, we had also collected swabs by sending a heli-copter to remote places with our medical team for the management.'* -Male, 20s, rural, provincial hospital

**3. Positive lessons.** Almost all the participants shared that the solutions they identified amid the challenges allowed them to appreciate the positive lessons from their experiences. Most of them could internalize the lessons learned, actively plan for future activities, and provide suggestions to pave the way for sustainability. The positive lessons they expressed included professional and personal lessons.

*3.1 Professional lessons.* The professional lessons shared by most health professionals included their reflections on the importance of teamwork, self-sustainability, and evidence-based practice.

***3.1.1 Self-sustainability.*** Almost everyone believed that such positive lessons will help them, in the long run, to lay out sustainable solutions and better preparedness for future crises. Almost all the health workers reflected that before the pandemic, the health sector generally depended on importing healthcare products and equipment. They felt that the pandemic taught them the importance of building local capacity and promoting local production for self-sustainability. They were also able to appreciate the existence of local opportunities that they did not realize before.

> *'I think COVID-19 highlighted the need for us to embrace a self-sustainable model as we did not have enough capacity or resources. Thankfully, Nepal has now started producing its own masks, PPE sets, and sanitizers. We have now even begun to manufacture or modify the venti-lators locally in Nepal. I think down the road; it will benefit us if we can learn the lessons and move forward more constructively.'* -Male, 30s, urban, central hospital

***3.1.2 Teamwork.*** The majority of the frontline health workers across all types of health facilities highlighted the importance of teamwork, indicating that they would not have been successful in dealing with the challenging situations of COVID-19 without good teamwork. Many expressed that gaining this insight during the crisis would enable them to continue strengthening team building in the coming days.

> *'Since the 2015 earthquake, this was the first disaster that we experienced in our life. It taught us lessons about infectious disease management, human resource mobilization, the infrastruc-ture needed, patient care, and overall about being well-prepared in advance. It taught us about ways to overcome our psychological stress.'* -Female, 50s, urban, and central hospital

> *'The major reason behind the efficient work here at our institution is teamwork. Most of the members in management here are pretty young, mostly less than 40 years of age, so they are*

*energetic, dedicated, and ready to work harder. That's why it was possible for us to manage effectively and efficiently.'*- Male, 30s, rural, provincial hospital

**3.1.3 Policy direction and research.** Many frontline health workers realized the importance of having policies and evidence-based protocols in place. They realized and also suggested the need for strengthening policies for pandemic preparedness and the need for data-driven decision-making. Some also highlighted that we must learn from the mistakes of this pandemic for better preparedness in the future. Some outlined the solutions for future preparedness, including improving contingency plans, risk communications, task shifting and better mobilizing human resources for health from different sectors.

*'During the earlier stage, there were no evidence-based protocols to follow. We were worried about the wrong treatment approaches. It would have been better if all the experts in the health sector had come together sooner to develop standard protocols at an early stage. We have protocols in place now, but that would have guided us better in the initial days when we all were lost.'*—Male, 40s, urban, central hospital

*'Future directions for prevention and treatment of this kind of health emergency should be managed by learning from the mistakes that we made in this pandemic. Research and public health strengthening within the Health Emergency Operation Center should be emphasized. We need a strong team including public health professionals and research, funding of epidemic, formulation of contingency plan, formation of the instant command center, and improving risk communication.'*—Male, 30s, urban, central hospital

**3.2 Personal lessons.** Many health workers were able to introspect at a personal level and articulate how the experiences have helped them. They shared how they realized their potential and how the experiences would help them in the future.

**3.2.1 Self-reflection.** Many expressed they did not realize they had the potential within themselves to fight the challenges until they overcame them.

*'We faced a situation where we must protect ourselves as well as our patients. We worked 24/7; we stayed away from family and had a hectic work schedule, which made us realize that we have been mentally and physically strong. We learned to handle pandemics in more ways than we knew we were capable of.'*–Female, 40s, urban, central hospital

*'I am taking this pandemic as an opportunity to work on my morale and confidence, which are higher after my COVID-19 duties, and I feel ready for this kind of challenge ahead.'*- Female, 40s, urban, provincial hospital

**3.2.2 Lesson from the patients.** Many also reflected on how their experience with treating the patients helped them to realize the essence of quality patient care, empathy, and compassion.

*'There was a female patient who tested positive for COVID-19 during the screening process when returning from India. Her brother-in-law was also positive. Her brother-in-law became negative after one week and went home. However, her PCR was positive even on the 4th attempt. She had developed suicidal thoughts too. So, we supported her by playing music, dancing, and sessions of counseling to make her mentally stable. We were able to make her mentally strong before she left. It gave me satisfaction, and I realized the importance of providing timely psychosocial support.'* -Female, 40s, urban, central hospital

*'The first COVID-19 patients in our hospital were three people who were stranded on the Nepal-India border due to a lockdown. Initially, they were furious as they had no idea why they were brought to the hospital. As the communication wasn't clear, keeping them in isolation wasn't easy. They were crying because they couldn't contact their family and weren't cooperating with us in our treatment plan. Gradually, our communication kept improving. Eventually, they started trusting us more and felt sorry for their earlier behavior toward us. They were saying, "come to visit us in our home. We would like to see your face and ask for your forgiveness. Until you forgive us, God will not forgive us." It allowed me to see through their vulnerabilities and the value of kindness.'*-Female, 40s, urban, provincial hospital

**4. Motivations.** The positive deviants mostly perform the way they do because they feel motivated. Most health workers shared that the driving force behind their solution-seeking behaviors was mostly intrinsic in nature. However, extrinsic factors also seemed to have played crucial roles for most health workers.

*4.1 Intrinsic motivation.* Intrinsic motivation is the underlying desire or willingness to work [27]. It comes from within when you are intrinsically motivated; you engage in an activity solely because you enjoy it and get personal satisfaction [27, 28]. Almost all health workers highlighted the motivational factors behind their desire to work even during the challenging situation of COVID-19. Some frontline health workers revealed that their profession was the major motivation factor to work and serve the country during this pandemic, while some perceived it as an opportunity to work harder. For some, their intrinsic empathy and compassion toward their patients motivated them to provide better care.

*'In our team, no one was demotivated. Instead, what we said was–we have completed a one-year fellowship in COVID-19.'*- Male, 40s, urban, central hospital

*'I am also motivated by the work of my colleagues; some of them are old in age, and some of them left their small children at home to come to work in such challenging times. Their willingness to work motivates me to work even more. Moreover, after seeing the staff working happily even in the heavily infected regions, I feel like, "They are working happily even in such challenging places, then why shouldn't I?".*-Female,20s, urban, central hospital

*'I once saw a patient who was a migrant worker who had returned from a foreign country. He had no money and looked tired, unclean, and unhappy. Despite the need to maintain physical distance, I couldn't resist getting closer to the patient and comforting him. I helped clean the patient, shave his beard, and bathe him. I still remember that look in his eyes after he looked fresh, clean, and motivated. And I believe this is the core principle of nursing care. We must take care of our patients from all aspects'*-Female, 50s, urban, and central hospital

*4.2 Extrinsic motivation.* Extrinsic motivation arises from outside and is enhanced by the work environment. High motivation can lead to better performance and high levels of satisfaction among workers, a better understanding of health workers' motivation are essential to better functioning of healthcare delivery systems [29, 30]. Most health workers expressed that motivation from others helped them perform better.

***4.2.1 Appreciations and acknowledgments.*** Most health workers across all types of health facilities felt motivated when they were appreciated by the patients, government, and media.

*'Once, we had a zoom meeting of medical superintendents with the director and provincial ministers. After sharing all the preparedness plans and experiences of our remote district, they*

*were impressed and said, "we have to learn from you; we should not always complain about the issues; instead, we have to take our initiatives." Their acknowledgment motivated me to do even better.'-Male, 20s, rural, district hospital*

*'This district's senior journalists from a famous news portal were also admitted to our isolation wards due to COVID-19. They were impressed by our work and communicated that in the news. It helped people have a positive impression of our work. We were encouraged by that.'-Female, 40s, urban, provincial hospital*

Some of the frontline health workers felt they were not the only ones who should be appreciated as there were other human resources such as technicians, cleaners, drivers, and other hospital team members who also had significant roles in managing the pandemic, and they were left unrecognized.

*'I think hospital cleaners are also unsung heroes. Despite their obvious effort in waste management, disposal, and patient transportation, they are not acknowledged enough. During the peak of the pandemic, many staff was infected, and we faced a severe shortage of human resources. The new cleaning staff had to support clinical staff in ICU even without knowing basic safety measures. They had to work even with basic orientation and were also at higher risk.'—Female, 50s, urban, central hospital*

**4.2.2 Family and social support.** Many expressed that the encouragement and warmth they received from their family and friends kept them going.

*'I was extremely scared of COVID-19 after going through the news around the world, even though it was yet to be reported in Nepal. My family members were very supportive. They encouraged me to continue to work and prepare myself to help COVID-19 patients. They said, "leaving the job is not a solution/option, do your duty carefully." I got moral strength from my family.'—Female, 20s, urban, central hospital*

*'Initially, I felt like leaving the job during this pandemic due to fear and uncertainties. However, my family was positive regarding my work. They motivated and encouraged me to perform my duty with adequate PPE.'-Female, 20s, rural, provincial hospital*

**4.2.3 Psychosocial support for peers.** Many expressed that the psychosocial support provided by their work colleagues kept them going. Many health workers came up with their solutions for peer support and designed different psychosocial initiatives for their mental well-being.

*'In order to support our colleagues and motivate them in this difficult time, we started the webinar series on "Positive Vibes during COVID-19". These webinars aimed to disseminate and share scientific knowledge, connect all the anesthesiologists across Nepal, and provide them with a platform to share concerns and knowledge. It was also a platform to provide moral support to each other. This webinar was helpful to relieve the stress among health workers across the different parts of the country such as Jumla, Surkhet, Dharan, and other places.'—Male, 30s, urban, central hospital*

*'We realized that it was important to boost the morale and confidence of the hospital staff. So, we conducted motivational psychological classes on a regular basis for the hospital staff to make them psychologically strong to perform their work.'—Female, 40s, urban, central hospital*

***4.2.4 Support from higher authorities.*** Many highlighted that support from the higher authorities, such as the government or the hospital authorities, made a positive impact and motivated them to work better despite the difficult circumstances.

*'I have worked closely with the government before as well. But I hadn't witnessed such coordination, communication, and team spirit as I witnessed this time. In the early days of COVID-19, the army, police, local government, and everyone worked together. I think it taught us that we should work like this in other crises as well.'*- Male, 30s, urban, central hospital

*'We were not in a position to do everything on our own. The provincial government helped us the most. The provincial government helped us directly in establishing PCR labs and ICU beds and mobilizing human resources. The expenses of mobilizing human resources were also covered by the province. Even the chief minister visited our hospital twice and assured us of any kind of help if needed. This pandemic also allowed the opportunity for the provincial and local governments to show their efficiency in managing pandemics.'* Male, 40s, rural, provincial hospital

## Discussion

This study explored the positive deviance traits among the COVID-19 frontline health workers in Nepal. The positive traits identified included overcoming challenges, finding innovative solutions, learning positive lessons from the crisis, and motivation behind their actions. The key learnings from this study include self-sustainability and timely actions, which can be incorporated in future planning of pandemics and other disasters. Other learnings include efficient management of local resources, which could be incorporated into regular healthcare management practices. The findings also highlighted the policy and implementation gaps in the readiness for the pandemic and the risks the frontline health workers had to go through in resource constraint settings.

The study findings depict that the frontline health workers were able to find solutions to the challenges they were facing while dealing with COVID-19. Finding solutions even in the most difficult circumstances is considered an important trait of positive deviance [29]. The health workers in this study were able to navigate through solutions and overcome their personal and professional challenges. The personal challenges shared by the frontline health workers in this study were comparable to the ones commonly experienced by the frontline health workers in Karachi, Pakistan [30]. Such challenges included stress and fear of the unknown, increased work hours, and being away from family members [30]. Studies have also identified increased anxiety and mental health problems among health workers during COVID-19 [31, 32]. The prevalence of anxiety and depression symptoms among healthcare workers in Nepal during the pandemic ranged between 41.9% to 46.95% and 37.5% to 41.31%, respectively [20, 32]. Similar to the experiences of stigma in this study, a previous study from Nepal reported that frontline health workers were six times more likely to be stigmatized than non-frontline healthcare workers [33].

The professional challenges faced by the health workers in this study were more specific to resource constraint settings with poorer health systems in place, which were more severe in remote facilities such as primary health centers and district hospitals. However, the bigger hospitals at provincial and central levels also faced resource constraints such as inadequate PPE, laboratory equipment, and human resources. While all the countries faced the crisis of inadequate resources, the burden was much higher in LMICs and particularly in poorer communities [32]. The frontline health workers have been playing a key role and are the pillars of the

health system of every country. Globally, frontline health workers have tackled the pandemic courageously amid the high burnout rate [31]. Despite the challenges, the frontline health workers have shown resilience in tackling stress in this study, similar to other studies conducted among health workers in Nepal [32, 34].

Finding low-cost and self-sustaining innovations and solutions were positive traits identified in this study, which is also an essential trait of positive deviance [35]. The local innovations may not always be of high technology. They could be frugal and fragile in nature, and such low-cost solutions are often the most critical solutions in times of need. However, the majority often tend to overlook the opportunities around their surroundings. Whereas the positive deviants try to actively seek solutions through uncommon practices [36]. Examples of using locally available materials such as plastics or raincoats as an alternative to PPE in this study may be seen as a risky step in general circumstances or high-resource settings. The frugal innovations in this study were temporary measures when safety issues had been largely not proven. However, in times of crisis, such as the early phases of pandemics when vaccines and the proper PPEs were unavailable, such actions could have protected the health workers exposed to such high risks. A review focusing on nurses' role as positive deviants indicated that decision-making in clinical settings often demands risk-taking and deviation from the regular standard practices [12]. A study about the US Army nurses explained how the nurses were able to provide the best care to the soldiers by adapting quickly to uncertainties of wartime situations deviating from the regular standard practices. The examples of nursing practices from wartime not only helped in more efficient patient management but also paved the way for better nursing practices in the future [37]. In this study, the health workers compared their sense of duty to the soldiers going to war. While the examples are not directly comparable, the findings highlight how the important traits of taking risks and innovating local solutions can tackle challenges during times of great uncertainty, such as pandemics and other disasters.

The health workers in this study not only successfully acted on the solutions but were also able to articulate the positive lessons they learned from those experiences. The ability to step back, reflect, and learn based on their experiences is also considered an important trait among positive deviants [12]. The positive deviants are known for finding opportunities and applying positive lessons amid constraints [36]. It allows their experiences and solutions to evolve better and become sustainable. The key lessons highlighted by the health workers in this study included the importance of self-sustainability and how crises and lack of resources can often push people to find local solutions. The health workers in this study provided insights on learning from their personal experiences as an opportunity to enhance their skills and morale. They also highlighted the value of teamwork as an important lesson they learned while finding solutions. Such traits are positively contagious and capable of inspiring and motivating others in a team in different healthcare settings [11]. A study on nurses described that the colleagues identified the positive deviant nurses as those who could bring everyone together as a team and were insightful and resourceful to bring positive changes [38].

The health workers in this study were also able to recommend realistic suggestions based on their lessons learned to tackle such pandemics and future crises of similar scales. The key suggestions included the need for more appreciation and motivation of all cadres of the health workforce, strengthening policies for pandemic preparedness, and the need for data-driven decision-making. A qualitative study in Pakistan also highlighted the need for strengthening leadership, a safer environment for health workers, and supportive and motivating working environments [30]. The current pandemic has highlighted the gaps in the public health system globally and even worse in low-resource settings where the applicable evidence-based guidelines and resources are not in place [39]. Moreover, the health workers highlighted the need for timely appreciation and acknowledgment which indeed could help sustain the progress

laid out by the positive deviants. A study on effective communication in family planning between Indonesian nurses and their patients also identified appreciation from the patients and peers as important factor for the positive deviants in sustaining the success of their work [40]. The suggestions based on the lessons learned by the positive deviants in this study provide insights on identifying sustainable solutions for Nepal not only to fight pandemics but also to strengthen the health system in general.

This study explored the different factors that motivated the participants to perform the way they did despite the challenging circumstances. The motivations behind their actions included both intrinsic and extrinsic factors. The intrinsic motivations reflected in this study were deeply rooted in their sense of duty, compassion, and values. Such intrinsic qualities have been identified in many studies focusing on the positive deviants among health workers, particularly among nurses [12, 40, 41]. Studies have indicated that positive deviant health workers are often driven by honorable intentions and values and are most capable of going to lengths for the sake of their patients, even if it means bending certain rules or taking risks [41–43]. In this study, too, there were examples of nurses willing to take personal risks to reach out to patients to provide compassionate care. Certain rules that were introduced in the earlier days of the pandemic, such as maintaining physical distance from the patients while treating them, were not entirely evidence-based and scientific. While bending such rules is often considered a risky step, positive deviants are more intuitive and can selectively bend certain rules for the greater good [43]. While such deviant behaviors are not often appreciated by the management in general, it is important to be open and allow flexibility and modification at times of pandemic and crisis [12].

Extrinsic motivation, such as appreciation and support, also play crucial roles in keeping the positive deviants driven. High external motivation has proved to result in better performance and high levels of satisfaction among workers and are essential to the better functioning of healthcare delivery systems [28, 29]. Similar to this study, family support, patients' appreciation, feedback from colleagues, and rewards have been identified as important external motivating factors for positive deviants to succeed [40]. Understanding the extrinsic motivating factors among the positive deviants provides insights into considering enabling factors while designing positive deviance interventions on a larger scale. It shall help scale up and sustain the progress initiated by the positive deviants.

## Limitations

This study has certain limitations. Firstly, the health workers' experiences in low-resource settings may not be generalized. Secondly, we may not have captured all the positive deviance traits among frontline health workers representing different cadres of health workers in Nepal. The majority of the health workers represented doctors and nurses, and other cadres of health workers, such as laboratory scientists and other paramedics, were under-represented. Despite the efforts to collect data from different cadres of health workers representing different regions and types of facilities in Nepal, this study was conducted amid the peak of the pandemic when the health workers were extremely busy. Thirdly, though efforts were made to minimize different types of biases, the respondents' biases, such as recall and social desirability, cannot be completely ruled out. Fourthly, this study has only focused on the initial two steps of the positive deviance approach, and further stages of testing and implementing the positive deviance traits among a larger population are necessary to generate better evidence. Lastly, the examples of their actions were unique to the earlier days of the pandemic when the world still lacked adequate evidence, which may not always be replicable. Despite the limitations, this study provides insights into the positive deviants among frontline health workers during the COVID-19

pandemic. The qualitative findings from the perspectives of positive deviants, such as in this study, are considered to be the initial steps in designing positive deviance interventions at a larger scale to assess the positive outcomes both qualitatively and quantitatively [30].

### Learnings for policy

The findings serve as a reminder to policymakers, implementers, and practitioners to be better prepared to avoid a similar crisis and also encourage positive deviance practices in the future. The study also reflected better coordination with the local and provincial government, particularly in remote districts, indicating opportunities for better public health management in Nepal. The findings based on the positive lessons in this study could encourage future positive deviance interventions for frontline health workers in low resource settings.

## Conclusion

The positive deviance traits identified in this study included finding solutions, innovations, learning positive lessons, and providing realistic suggestions and their underlying motivations. The findings of this study provide insights into how positive deviance can help health workers working in low-resource settings and highlights the need for self-sustaining solutions for such settings. However, in the future, larger studies are warranted to explore deeper into positive deviance and look into the effects of positive deviance in bringing positive outcomes during the pandemic.

## Supporting information

**S1 Table. Interview guide.**
(DOCX)

**S2 Table. COREQ checklists.**
(PDF)

## Author Contributions

**Conceptualization:** Rolina Dhital.

**Data curation:** Rolina Dhital, Madhusudan Subedi, Pawan Kumar Hamal, Carmina Shrestha.

**Formal analysis:** Rolina Dhital, Madhusudan Subedi, Carmina Shrestha, Sandesh Bhusal, Reshika Rimal, Lovin Gopali, Richa Shah.

**Investigation:** Rolina Dhital, Madhusudan Subedi, Pawan Kumar Hamal, Carmina Shrestha, Sandesh Bhusal, Reshika Rimal, Lovin Gopali.

**Methodology:** Rolina Dhital, Madhusudan Subedi, Pawan Kumar Hamal, Carmina Shrestha, Sandesh Bhusal, Reshika Rimal, Lovin Gopali, Richa Shah.

**Project administration:** Rolina Dhital, Madhusudan Subedi, Pawan Kumar Hamal, Carmina Shrestha.

**Resources:** Rolina Dhital, Madhusudan Subedi, Pawan Kumar Hamal, Sandesh Bhusal, Reshika Rimal, Lovin Gopali, Richa Shah.

**Software:** Rolina Dhital, Pawan Kumar Hamal, Carmina Shrestha, Sandesh Bhusal, Reshika Rimal, Lovin Gopali, Richa Shah.

**Supervision:** Rolina Dhital, Madhusudan Subedi.

**Validation:** Rolina Dhital, Madhusudan Subedi, Pawan Kumar Hamal, Sandesh Bhusal.

**Visualization:** Rolina Dhital, Madhusudan Subedi, Pawan Kumar Hamal.

**Writing – original draft:** Rolina Dhital.

**Writing – review & editing:** Rolina Dhital, Madhusudan Subedi, Pawan Kumar Hamal, Carmina Shrestha, Sandesh Bhusal, Reshika Rimal, Lovin Gopali, Richa Shah.

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
