## [Decision Letter · Decision Letter 0]

18 Jul 2022

PGPH-D-22-00902

How positive deviants helped in fighting early phase of COVID-19 pandemic? A qualitative study exploring roles of front-line health workers in Nepal

Dear Dr. Dhital,

Thank you for submitting your manuscript to PLOS Global Public Health. After careful consideration, we feel that it has merit but does not fully meet PLOS Global Public Health’s publication criteria as it currently stands. Therefore, we invite you to submit a revised version of the manuscript that addresses the points raised during the review process.

I agree with the reviewers that there is a lot of rich and useful insights in your manuscript. Their credibility will be enhanced and your arguments are likely to be strengthened by revisiting the methods as well as analysis in your manuscript, as specifically highlighted by Reviewer 5. I look forward to reading your revised manuscript once you have had an opportunity to address all the review comments.

We look forward to receiving your revised manuscript.

Kind regards,

Syed Shahid Abbas, MBBS, MPH, Ph.D.

Academic Editor

Journal Requirements:

1. In the online submission form, you indicated that "Data will be made available upon reasonable request as the data is not entirely anonymous.". All PLOS journals now require all data underlying the findings described in their manuscript to be freely available to other researchers, either 1. In a public repository, 2. Within the manuscript itself, or 3. Uploaded as supplementary information.

2. We have noticed that you have uploaded Supporting Information files, but you have not included a list of legends. Please add a full list of legends for your Supporting Information files after the references list. 

Additional Editor Comments (if provided):

Reviewers' comments:

Reviewer's Responses to Questions

**Comments to the Author**

1. Does this manuscript meet PLOS Global Public Health’s publication criteria? Is the manuscript technically sound, and do the data support the conclusions? The manuscript must describe methodologically and ethically rigorous research with conclusions that are appropriately drawn based on the data presented.

Reviewer #1: Yes

Reviewer #2: Yes

Reviewer #3: Yes

Reviewer #4: Yes

Reviewer #5: Yes

Reviewer #6: Yes

2. Has the statistical analysis been performed appropriately and rigorously?

Reviewer #1: No

Reviewer #2: N/A

Reviewer #3: N/A

Reviewer #4: Yes

Reviewer #5: N/A

Reviewer #6: N/A

3. Have the authors made all data underlying the findings in their manuscript fully available (please refer to the Data Availability Statement at the start of the manuscript PDF file)?

Reviewer #1: Yes

Reviewer #2: No

Reviewer #3: Yes

Reviewer #4: Yes

Reviewer #5: No

Reviewer #6: Yes

4. Is the manuscript presented in an intelligible fashion and written in standard English?

Reviewer #1: Yes

Reviewer #2: Yes

Reviewer #3: No

Reviewer #4: Yes

Reviewer #5: No

Reviewer #6: Yes

5. Review Comments to the Author

Reviewer #1: Dear Authors, it was a pleasure reading your manuscript. Congratulations on your findings and using the innovative method to look at the perspectives of frontline HCWs from a different angle. Although I am not a core qualitative researcher nor a social scientist, who could have given more detailed feedback, as a general medical/public health researcher, I have the following comments for your consideration:

Abstract: Please elaborate results section. Introduction and method section could be reduced. Mention key sub themes, important traits, etc

Methods/Results:

- Under types hospital, readers would like to see the difference in findings between public and private healthcare facilities. Out of the selected health facilities, how many of them were recognized as COVID-19 designated HCFs by the government? Was there any difference in findings between designated and non-designated HCFs?

Results:

- Table 2: Please write the legend in full. What do these codes, categories and themes reflect or relate to? COVID-19 response in resource-poor healthcare settings?

- It would be better to give numbers to sub themes (for e.g., 1.1 personal challenges, 1.2 professional challenges) for easier reading and understanding.

Discussion:

- It would be helpful to discuss the differences of findings (motivation, challenges/experiences, recommendations) across health profession/cadres, types of HCFs (public vs private), and rural/urban settings.

General: Although it is a qualitative study conducted among 17 participants, readers would like to see the deviation of study findings across different cadres/profession, types of hospital, rural/urban setting, etc. measured in proportion or percentage. Please review statistical aspects.

Reviewer #2: Comments:

Major comments:

1. No clear mention of how those 17 participants were selected for the interview

2. Pertinent and relevant discussion of the author's research finding is mentioned in the discussion; however, it is recommended to discuss more the psychological impact among health care workers from Nepal in addition to the global impact authors have mentioned in their second paragraph of discussion, including findings from some pertinent literature https://pubmed.ncbi.nlm.nih.gov/34345537/, https://pubmed.ncbi.nlm.nih.gov/33566863/, etc.

Minor comments:

1. The introduction is too lengthy and contains very general information, which may not be necessary for the reader of scientific journal articles. Authors may keep some of that information in the discussion section instead

2. Unsure about the actual date versus typo error? "The interviews were conducted between 8 December November 2020 and 16 March 2021 and there were no repeat interviews."

3. Poof reading and minor edits of English and language-related issues are recommended

Reviewer #3: 1 Introduction

• Line 56-57: Reference required

• Line 58-59: Reference required

• Line 74-87: Reference required

• Line 77-81: Reference required

2 Materials and Methods

a) Steps of positive deviance approach:

• Please paraphrase and shorten the paragraphs. It is very confusing at the moment. The description/steps of the

positive deviance initiative should be at the beginning to introduce the concept and the differences followed by

it.

b) Data Collection:

• Please elaborate how the privacy and confidentiality was maintained.

• Please write the exact data collection period. Is it November or December 2020?

• Did you obtain a written informed consent or a verbal informed consent? Specify.

3 Results

• The interpretation in regards to the themes and subthemes are inadequate. Some subthemes directly consist of

the quotation without any interpretation and this is not correct. Quotations are there for supporting your

interpretation not a replacement for it.

• The subtheme- Finding solutions within the theme overcoming Challenges more or less corresponds to

innovative solutions. Why not you name this theme as Challenges and then only present personal and

professional challenges.

• May be you can merge subthemes Practice based innovations and technological innovations/Practices .

• You have not written anything on the subthemes: Lessons from organization and lessons from government.

Please check.

• Line 297-301: What does 5 PPE mean? Is it the PPE set or a particular PPE?

Overall: Please cut down the use of the article “The”

Use gender neutral terms as far as possible. E.g Replace manpower with human resource

Check the grammar in the whole manuscript.

Use active voice rather than passive specifically in the methods section

Reviewer #4: The study is usefult to gain insights about positve deviants that front health workers experienced during the early phase of COVID-19 in Nepal. The authors could benefit from the following suggestions:

1. Methods, Line 158-159, There seem to be an error in reporting the duration of data collection. Please correct it

2. Methods, Line 161-164, Was the consent taken in written form or video-recorded during the interview?

3. Results, Line 180-182, Can the number of participants be disaggregated by gender and also by province? Was there any specific reason for choosing more participants from Mountain than plain? I was particularly concerened about why health workers like laboratory personnel were not included in the study.

4. Results, Line 189, Code of Theme 5 (Recommendations) seems to be missing.

5. Discussion, Line 642-657, This paragraph is more about explaining what intrinsic and extrinsic motivation are. I would like to suggest authors to relate to the study findings, describe it or compare it with other settings and draw implications from it.

6. Please correct some typos such as 'FInding solutions' (Line 189)

Reviewer #5: This is a good and relevant study and my hearty congratulations to the authors for working on an interesting topic. I feel that the study is interesting, but the paper needs more details and revisions for it to be well interpreted by an external reader. Sharing some suggestions below.

I know this list might look a bit overwhelming at first (my very sincere apologies) but hope it is helpful in moving forward.

Introduction

• The Positive Deviance approach needs to be defined clearly, using references in the first sentence. Sentence 42-43 may need to be tweaked since PD as a practical strategy enables social and behavioral changes through looking at outliers (does not address social and behavioural changes though). I think in this paper, PD is used to identify ‘champions of change’, outliers who succeeded despites odds during a very challenging situation. This paper has a good overview of various concepts of definitions of PD-Herington MJ, van de Fliert E. Positive Deviance in theory and practice: a conceptual review. Deviant Behavior. 2018;39(5):664–78.

• Line 52- few more paper references can be used rather than only one. Particularly papers that have used PD approaches to model frontline health worker behaviours previously (rather than those that looked at community behaviours)

• Line 53- tweak. PD approaches can’t be used to address the pandemic. Restate-PD approaches haven’t been used to study frontline workers in the context of COVID-19. I like lines 82-83 that convey this point better.

• Lines 81-82 have good arguments for the selection of positive deviants.

• Lines 87-88: it could be interpreted as the frontline workers interviewed were diagnosed with COVID-19. Aim of the study - suggestion- explore the experiences of positively deviant frontline workers during early days of the pandemic

Methods

• Identification and selection of participants has been clearly written, data collection processes are also very well defined.

• It would be good to have a small box/table on the themes covered during the interviews.

• Data analysis lines 172-176 are a little confusing for an external reader. Was the coding tree created before the reading of the transcripts? (from the interview guide)

• Table 2 with codes can come into the analysis section as it usual practice in qualitative papers.

• I am a little confused about the methods followed. Lines 111 -115 state that the steps defined by the positive deviance initiative were followed for the research. But the findings are not reported in terms of these steps. I see step 2 (identification of positive deviants) and a bit of step 3(discovering successful behaviours). I don’t see the other steps.

• Table 1 could have age, gender and other details- as usual demographic tables in qualitative papers.

Findings

• This section may need a lot of rewriting. It has many quotes and very rich, beautiful quotes- but these need to get tied into a narrative better. It is important to present quotes from the ground, but much of qualitative writing is about interpreting and weaving a story around these quotes. Some of this story is in the discussion section. Overall, the section has many quotes with very little reflection from the authors side on what these quotes mean or how can these fit together to tell a story. Hence, I would suggest major rewriting.

• Lines 195 onwards-too many details on the challenges faced.

• Maybe theme 1 Overcoming challenges section on “Finding solutions” can be merged with theme 2 “innovative solutions”. These two themes seem very similar.

• The theme on innovative solutions (line 322) feels like the meat of the paper to me. The findings here are fantastic and grounded, and we must tell the world about these!

• I do not understand the last theme on lessons at all. These is no text given by the authors to go along with the quotes. Much of the lessons seem like a reiteration of the overcoming challenges section in a different way.

• One way of tying these themes together is as below (only a suggestion and there can be other ways). First, you can describe (briefly) some of the challenges faced by frontline workers. Then a long description of how positive deviants attempted to overcome these challenges and the lessons they learnt during this process. This is the meat of the paper and used reflections/thoughts/ data displays like tables, quotes, even figures. Last, a section on what different factors motivated positive deviants to behave differently from others and find innovative solutions when others didn’t go so.

• One data analysis book that might be useful is Miles, M.B., Huberman, A.M., 1992. Qualitative Data Analysis.

• Also it might help the authors to refer to how quotes are written in qualitative papers. Usually age and gender are used in quotes to give a better sense of the respondent profile. One cannot make out from these quotes the diversity in participants.

Discussion

• Has good referencing and the authors have attempted to synthesize lessons. Some of the narrative here- example lines 636-640, lines 658-666 all seem like findings rather than discussion. In the discussion, I would be more interested in reflection about what these findings mean for future pandemics, an action box for the way forward, etc.

Reviewer #6: Thank you for the opportunity to review this paper – it was very interesting to read about the traits among positive deviants and their approaches used in managing the COVID-19 pandemic in low resource settings.

The following are my suggested revisions/questions to clarify in the report –

• In the background, I felt it would be useful to provide some examples of how positive deviance was used within the other public health programs mentioned (e.g. maternal and child health)

• With regard to HCWs during the pandemic, it is mentioned that “Better knowledge and skills, better planning and management, etc. were some of the key qualities among these health workers which enabled them to perform better” – is there a reference for this? Did these findings also inform the conceptual framework/ interview guides used in the study?

• In the methodology, it seems to me that the Positive Deviance Initiative is the approach that this study follows – I felt a clearer justification for using this approach was needed, and maybe the differences between the various approaches could be taken up in the discussion section.

• In the results, would it be possible to provide any further demographic details on the study participants (e.g. age, marital status, did they themselves have any medical conditions, etc.) and also how these could have affected the study findings.

• I also felt further synthesising of the findings presented in the quotes were needed for some of the themes and sub-themes reported (e.g. stigma, changing working patterns, psychosocial support for peers, etc.)

Many thanks for considering my comments.

6. PLOS authors have the option to publish the peer review history of their article (what does this mean?). If published, this will include your full peer review and any attached files.

**Do you want your identity to be public for this peer review?** For information about this choice, including consent withdrawal, please see our Privacy Policy.

Reviewer #1: No

Reviewer #2: No

Reviewer #3: No

Reviewer #4: No

Reviewer #5: No

Reviewer #6: No

---

## [Decision Letter · Decision Letter 1]

6 Oct 2022

PGPH-D-22-00902R1

How positive deviants helped in fighting the early phase of COVID-19 pandemic? A qualitative study exploring the roles of front-line health workers in Nepal

Dear Dr. Dhital,

Thank you for submitting your manuscript to PLOS Global Public Health. After careful consideration, we feel that it has merit but does not fully meet PLOS Global Public Health’s publication criteria as it currently stands. Therefore, we invite you to submit a revised version of the manuscript that addresses the points raised during the review process.

I will encourage you to reflect on the comments from Reviewer 5. Please feel free to draw upon the expertise of colleagues with writing experience to help further develop your synthesis of the quotations and raw data. In addition, you might also find it helpful to engage a copy editor to proof read the text.

We look forward to receiving your revised manuscript.

Kind regards,

Syed Shahid Abbas, MBBS, MPH, Ph.D.

Academic Editor

Journal Requirements:

Additional Editor Comments (if provided):

Reviewers' comments:

Reviewer's Responses to Questions

**Comments to the Author**

1. If the authors have adequately addressed your comments raised in a previous round of review and you feel that this manuscript is now acceptable for publication, you may indicate that here to bypass the “Comments to the Author” section, enter your conflict of interest statement in the “Confidential to Editor” section, and submit your "Accept" recommendation.

Reviewer #3: All comments have been addressed

Reviewer #4: All comments have been addressed

Reviewer #5: All comments have been addressed

Reviewer #6: All comments have been addressed

2. Does this manuscript meet PLOS Global Public Health’s publication criteria? Is the manuscript technically sound, and do the data support the conclusions? The manuscript must describe methodologically and ethically rigorous research with conclusions that are appropriately drawn based on the data presented.

Reviewer #3: Yes

Reviewer #4: Yes

Reviewer #5: Partly

Reviewer #6: Yes

3. Has the statistical analysis been performed appropriately and rigorously?

Reviewer #3: N/A

Reviewer #4: Yes

Reviewer #5: N/A

Reviewer #6: N/A

4. Have the authors made all data underlying the findings in their manuscript fully available (please refer to the Data Availability Statement at the start of the manuscript PDF file)?

Reviewer #3: Yes

Reviewer #4: Yes

Reviewer #5: No

Reviewer #6: Yes

5. Is the manuscript presented in an intelligible fashion and written in standard English?

Reviewer #3: Yes

Reviewer #4: Yes

Reviewer #5: No

Reviewer #6: Yes

6. Review Comments to the Author

Reviewer #3: The authors have addressed all the comments provided. There is no further comments from my side.

Reviewer #4: Authors have done a fair job addressing all the comments. Considering the knowledge contribution that the study brings to the COVID-19 pandemic and future outbreaks, I recommend the manuscript for publication.

Reviewer #5: Dear Authors

Thank you for your efforts in rewriting the article. I can understand that it is a lot of hard work bringing the manucript to this phase and it is not easy to work with reviewer comments. My apologies for the same.

I do still feel that some work is needed, particularly in synthesizing the data in the findings section. The quotes are still long. The discussion is also long winding and will benefit from some synthesis. I have given two sample papers below that might give a sense of how narratives can be written alongside quotes.

More detailed thoughts below

Abstract

Page 2- Line 19 Positive deviance is a concept that enables behavioral and social changes by learning 20 from the deviants who fare better than others under similar circumstances to find better solutions. Perhaps better to rephrase: Positive deviance is an approach wherein learnings from persons who fare better than their peers under similar circumstances are used to enable behavioural and social change.

Line 22 most socially and culturally contextual- fit into socio-cultural milieu, or fit the context well.

The use of positive deviance versus deviants needs to be checked throughout the paper.

Introduction

• Definition of positive deviance tweaked

• The use of positive deviance versus deviants needs to be checked throughout the paper.

• Lines 57-77 can be condensed to make the point that such studies are widely used and useful

• Line 78-79:rephrase. Given the usefulness of the positive deviance approach in understanding and enabling change in the health sector, the approach can be used in the context of COVID..etc ( References 14-16 are about positive deviants during COVID, so it might not be right to say that such studies have not been done at all)

• References 18 and 19 speak to Ebola and SARS in general, but do not convey the point that these did not affect Nepal. Need better references here.

Methods section

• Lines108-11 are very clear now.

• Liens 125 onwards is also very well written. The methods section is clear

Findings

• This section still needs work

• Quotes can be shortened and simplified. They are too long right now.

• I feel that the connecting narrative still needs to be reflected upon and strengthened. For instance, there is only 2 lines of narrative on stigma, but three quotes. Another example is the section on managing resources- there is only one paragraph of narrative followed by 2 full pages of quotes.

• I would suggest to build a stronger expanded narrative that synthesizes the information you got from the quotes, but keep only 1 or 2s quote that is most illustrative of the points being made.

• Some sample papers below that use quotes and case-illustrations alongside the narrative (just to give ideas on how the narrative around the quotes can be strengthened). The narrative has to weave the ideas in the quotes together, and 1-2 quotes or case-stories may be used to illustrate key ideas.

https://bmjopen.bmj.com/content/bmjopen/12/6/e050363.full.pdf

https://www.ijhpm.com/article_3942.html

Discussion- it would be nice to have one box that summarizes recommendations or learnings for policy from the study. The section can be shortened with clear suggestions or learnings. start the discussion with a clear summary of learnings (3-4 sentences). Usually the discussion section also has a small paragraph on the limitations of the study done, so it can be interpreted in context.

Reviewer #6: No further comments

7. PLOS authors have the option to publish the peer review history of their article (what does this mean?). If published, this will include your full peer review and any attached files.

**Do you want your identity to be public for this peer review?** For information about this choice, including consent withdrawal, please see our Privacy Policy.

Reviewer #3: No

Reviewer #4: No

Reviewer #5: No

Reviewer #6: No

---

## [Decision Letter · Decision Letter 2]

14 Feb 2023

How positive deviants helped in fighting the early phase of COVID-19 pandemic? A qualitative study exploring the roles of front-line health workers in Nepal

PGPH-D-22-00902R2

Dear Dr. Dhital,

We are pleased to inform you that your manuscript 'How positive deviants helped in fighting the early phase of COVID-19 pandemic? A qualitative study exploring the roles of front-line health workers in Nepal' has been provisionally accepted for publication in PLOS Global Public Health.

Best regards,

Julia Robinson

Executive Editor

Reviewer Comments (if any, and for reference):

Reviewer's Responses to Questions

**Comments to the Author**

1. If the authors have adequately addressed your comments raised in a previous round of review and you feel that this manuscript is now acceptable for publication, you may indicate that here to bypass the “Comments to the Author” section, enter your conflict of interest statement in the “Confidential to Editor” section, and submit your "Accept" recommendation.

Reviewer #1: All comments have been addressed

Reviewer #4: All comments have been addressed

Reviewer #6: All comments have been addressed

2. Does this manuscript meet PLOS Global Public Health’s publication criteria? Is the manuscript technically sound, and do the data support the conclusions? The manuscript must describe methodologically and ethically rigorous research with conclusions that are appropriately drawn based on the data presented.

Reviewer #1: Partly

Reviewer #4: Yes

Reviewer #6: Yes

3. Has the statistical analysis been performed appropriately and rigorously?

Reviewer #1: Yes

Reviewer #4: Yes

Reviewer #6: N/A

4. Have the authors made all data underlying the findings in their manuscript fully available (please refer to the Data Availability Statement at the start of the manuscript PDF file)?

Reviewer #1: Yes

Reviewer #4: Yes

Reviewer #6: Yes

5. Is the manuscript presented in an intelligible fashion and written in standard English?

Reviewer #1: Yes

Reviewer #4: Yes

Reviewer #6: Yes

6. Review Comments to the Author

Reviewer #1: Dear author(s), thank you for submitting the second revision of the paper.

The article has been much improved compared to its initial versions. However, there are still some areas of improvement, hence, suggesting a minor revision. Particularly, around discussion of the findings, the authors are suggested to explore what was already known and what this paper truly adds value in the field of health workforce's socio-behavioral changes through positive deviances.

Few comments:

1. The study revolves around 'role model' approach to enable behavioral and social change among at-risk frontline HCWs during pandemics such as COVID19. Please explore whether this approach has been recommended by other similar studies conducted during various points of the pandemic, and the impact.

2. Study setting, findings, discussion: Please mention the inclusion of hcws from public vs private sector too. Mention the differences identified, if any, in terms of acceptance to positive deviance approach, its likely impact, etc.

3. Challenges: Stigmatization of HCWs was at the highest level during the initial phase of COVID19 pandemic. Other studies have also reported similar stories, which could have discussed by this paper. For example, Aacharya, R. P., and Shah, A. (2020). Ethical dimensions of stigma and discrimination in Nepal during COVID-19 pandemic. Ethics Med. Public Health 14:100536. doi: 10.1016/j.jemep.2020.100536 . It mentions- Medical staff dealing with testing or working in hospitals, isolation wards or quarantine centres have been stigmatized as ‘possibly infected’ and even denied food and accommodation.

4. Challenges: On professional and personal challenges, technology, etc. there are other studies which interviewed higher number of HCWs from various sector and levels of health facilities. Such studies have identified a list of challenges and then recommended possible solutions, for example, Role model approach, incentivization of HCWs, evidence-based treatment of patients, peer-support, uniformity in treatment protocols and government notices, IPC or COVID19 committee formation at facility level, utlizing trainee workforce to meet high service demands, HWC's autonomy for decision-making particularly to address ethical dilemma, etc. which could be regarded as positive deviants too. This paper misses to discuss these points. An example of similar paper is, Participatory Approach to Develop Evidence-Based Clinical Ethics Guidelines for the Care of COVID-19 Patients: A Mixed Method Study From Nepal. Front. Public Health, 2022.

https://doi.org/10.3389/fpubh.2022.873881;

5. Resource related challenges and innovation: This study was conducted when the first wave of COVID19 was almost over (Nov2020), by that time, not only the country somehow managed necessary resources, many health development partners invested in capacity building of HCWs which included psychological support, awareness, etc. A study surveyed 100+ hospitals at the time of real chaos and identified the deficit in terms of health services, resources, training, etc. The same study also reported the disparity in terms of resource allocation. The findings from this particular study could have been discussed, maybe comparing different points of the pandemic and how HCWs perceived the development of pandemic, build their motivation, what positive deviants helped or did not help, etc. (Assessment of service availability and Infection prevention measures in hospitals of Nepal during the transition phase of COVID-19 case surge.

medRxiv 2020.05.13.20097675; doi: https://doi.org/10.1101/2020.05.13.20097675

.......Nasopharyngeal (NP)/throat swab kits were available in one-third (35/110), whereas viral transport media (VTM), portable fridge box, and refrigerator were available in one-fifth (20%) of hospitals. Only one hospital (large/tertiary) had a functional PCR machine. Except for General practitioners, other health cadres—crucial during pandemics, were low in number. On IPC measures, the supplies of simple face masks, gloves and hand sanitizers were adequate in the majority of hospitals, however, N95-respirators, Filter masks, and PPE-suits were grossly lacking. Government’s COVID-19 support was unevenly distributed across provinces; health facilities in Province 2, Gandaki, and Province 5 received fewer resources than others......).

6. Suggestion to expand discussion on technological innovation: See above (point 5). Other studies have identified the real lack of technology and resources which affected patient care during the first wave.

7. Teamwork, motivation of HCWs: See above (point 4).

Reviewer #4: The paper is in a good shape and looks fine for publication.

Reviewer #6: (No Response)

7. PLOS authors have the option to publish the peer review history of their article (what does this mean?). If published, this will include your full peer review and any attached files.

**Do you want your identity to be public for this peer review?** For information about this choice, including consent withdrawal, please see our Privacy Policy.

Reviewer #1: No

Reviewer #4: No

Reviewer #6: No
